# Perifocal Zone of Brain Gliomas: Application of Diffusion Kurtosis and Perfusion MRI Values for Tumor Invasion Border Determination

**DOI:** 10.3390/cancers15102760

**Published:** 2023-05-15

**Authors:** Natalia E. Zakharova, Artem I. Batalov, Eduard L. Pogosbekian, Ivan V. Chekhonin, Sergey A. Goryaynov, Andrey E. Bykanov, Anastasia N. Tyurina, Suzanna A. Galstyan, Pavel V. Nikitin, Lyudmila M. Fadeeva, Dmitry Yu. Usachev, Igor N. Pronin

**Affiliations:** Federal State Autonomous Institution “N.N. Burdenko National Medical Research Center of Neurosurgery” of the Ministry of Health of the Russian, 4th Tverskaya-Yamskaya Str. 16, Moscow 125047, Russia; nzakharova@nsi.ru (N.E.Z.); abatalov@nsi.ru (A.I.B.); pogos.ed@gmail.com (E.L.P.); sgoraynov@nsi.ru (S.A.G.); abykanov@nsi.ru (A.E.B.); aturina@nsi.ru (A.N.T.); sgalstyan@nsi.ru (S.A.G.); nikitinpaulv@yandex.ru (P.V.N.); lmf@nsi.ru (L.M.F.); dousachev@nsi.ru (D.Y.U.); pronin@nsi.ru (I.N.P.)

**Keywords:** high-grade gliomas, MRI biomarkers, diffusion-kurtosis MRI, ASL-perfusion

## Abstract

**Simple Summary:**

Preoperative determination of glioma invasion borders remains crucial for neuroradiology. Diffusion kurtosis imaging (DKI) is one of the promising tools as it reflects complex tissue microstructure. Pseudo-continuous arterial spin labeling (pCASL) perfusion is a reliable method to distinguish the most malignant tumor part. In 50 high-grade glioma patients, we demonstrated significant differences between the DKI values in peritumoral white matter (normal-appearing on conventional MRI) and unaffected contralateral hemisphere white matter, which may indicate possible infiltration of normal-appearing peritumoral white matter by glioma cells. The study demonstrated the presence of tumor cells within the edema zone in all gliomas. Tumor cells and tumor stem-like cells were detected in some samples of normal-appearing white matter surrounding glioblastomas. DKI and cerebral flow values showed correlations with quantitative neuropathological markers (proliferative or antiapoptotic activity) in gliomas. Thus, DKI can shed light on tumor-associated brain matter changes and is potentially capable of predicting morphological tumor properties.

**Abstract:**

(1) Purpose: To determine the borders of malignant gliomas with diffusion kurtosis and perfusion MRI biomarkers. (2) Methods: In 50 high-grade glioma patients, diffusion kurtosis and pseudo-continuous arterial spin labeling (pCASL) cerebral blood flow (CBF) values were determined in contrast-enhancing area, in perifocal infiltrative edema zone, in the normal-appearing peritumoral white matter of the affected cerebral hemisphere, and in the unaffected contralateral hemisphere. Neuronavigation-guided biopsy was performed from all affected hemisphere regions. (3) Results: We showed significant differences between the DKI values in normal-appearing peritumoral white matter and unaffected contralateral hemisphere white matter. We also established significant (*p* < 0.05) correlations of DKI with Ki-67 labeling index and Bcl-2 expression activity in highly perfused enhancing tumor core and in perifocal infiltrative edema zone. CBF correlated with Ki-67 LI in highly perfused enhancing tumor core. One hundred percent of perifocal infiltrative edema tissue samples contained tumor cells. All glioblastoma samples expressed CD133. In the glioblastoma group, several normal-appearing white matter specimens were infiltrated by tumor cells and expressed CD133. (4) Conclusions: DKI parameters reveal changes in brain microstructure invisible on conventional MRI, e.g., possible infiltration of normal-appearing peritumoral white matter by glioma cells. Our results may be useful for plotting individual tumor invasion maps for brain glioma surgery or radiotherapy planning.

## 1. Introduction

The overall survival rate of high-grade glioma patients, particularly those with glioblastomas, remains low despite combination therapy. This is facilitated by prognostically unfavorable infiltrative tumor growth patterns and diffuse involvement of brain structures.

High-grade gliomas are known for extension beyond the margin of enhancing tumor mass, usually removed during surgery. The glial tumor part with an intact blood-brain barrier (and thus non-enhancing) extends to the perifocal edema area [1]. Assessment of non-enhancing infiltrative tumor zone is mandatory in glioma imaging, especially in postoperative follow-up [2,3]. Still, changes caused by perifocal vasogenic edema tumor cell infiltration are not visualized on conventional MRI.

In autopsy cases, glioma tumor cells were detected remotely from the zone of MR signal changes on T2-weighted images [4]. Conventional MRI may underestimate glioma spread, which may lead to suboptimal tumor resection and outcome worsening [5]. Resection zone extension and even gross total resection of high-grade gliomas are the basic conditions of patient survival improvement [6,7].

Differentiation between purely vasogenic and glioma-infiltrated edema as well as detection of infiltrated zones in normal-appearing white matter could allow true tumor extension determination, surgery, and radiotherapy planning.

Different studies applied various MR sequences, among them yielding quantitative results (e.g., diffusion, relaxometric, perfusion) to determine real glioma spread. Diffusion-weighted imaging (DWI) could demonstrate an apparent diffusion coefficient gradient in peritumoral edema as a tumor infiltration marker based on cellularity [8]. Analysis of high-grade glioma MR relaxometry maps revealed significant peritumoral edema changes indicative of tumor invasion [3]. Elevation of cerebral blood flow (CBF) on CT- and MR-perfusion images is a consequence of tumor-induced neoangiogenesis. Previously, pseudo-continuous arterial spin labeling (pCASL) showed the capability to differentiate high- and low-grade gliomas [9,10]. Perfusion methods were used to study peritumoral edema [11] as well as being successfully employed to differentiate glioblastomas and metastases or glioblastomas and lymphomas [12,13,14].

Significant changes of fractional anisotropy (FA) derived from diffusion-tensor imaging (DTI) were detected in different studies not only in perifocal edema but also in the white matter beyond conventional MRI signal changes [15,16,17]. Nevertheless, some other studies did not reveal significant FA changes in normal-appearing white matter beyond the radiologically detected infiltrative edema zone [18,19]. At the same time, despite the high capabilities of modern MR scanners, all mentioned methods cannot unequivocally determine the true borders of brain matter infiltration. Thus, the interpretation of glioma imaging remains complex.

Diffusion kurtosis MRI (DKI) yields data concerning the non-Gaussian distribution of water diffusion, thus more precisely reflecting complex in vivo tissue microstructure [20,21,22,23,24]. According to various studies, DKI parameters are non-invasive biomarkers for glioma grading [25,26,27,28,29,30,31,32,33]. Mean kurtosis (MK), the most well-studied value in DKI, showed high precision in low- and high-grade glioma differentiation in many studies [27,34,35].

Former works applied multiparametric analysis for the prediction of glioblastoma recurrence direction and studied peritumoral infiltrative edema, which possesses a more complex structure compared to vasogenic edema typical for metastases and extra-axial tumors [36,37,38,39]. These works were aimed at obtaining pathophysiological information concerning neoangiogenesis, white matter integrity, and cellular proliferation on the basis of quantitative MR parameters in vivo to differentiate pure vasogenic edema from tumor-infiltrated vasogenic edema [33].

Several studies used biopsies to invigorate neuroimaging data. Guo J. et al., 2012, obtained MRI-guided needle biopsies from the perifocal infiltrative zone and normal-appearing peritumoral brain matter of grade 2–4 glioma patients [40]. MR-spectroscopy showed dependence between the Cho/NAA ratio and the presence of peritumoral infiltration. Cho/NAA ratio was proven to be a unique parameter to determine glial tumor borders.

Zetterling M. et al., 2016, compared MRI data and results of a histological study of en bloc resection-derived glial tumor samples and proved tumor infiltration presence beyond the abnormal MR signal zone [41].

During the study planning, we proposed that combined analysis of DKI and quantitative perfusion values could determine borders of non-enhancing high-grade glioma zones with the intact blood-brain barrier. Information concerning tumor spread will allow for the development of a personalized algorithm for surgical tumor removal with consequent radiotherapy and chemotherapy planning, as well as tumor course prognosis.

The aim of our research was to study high-grade glioma biomarkers based on DKI and perfusion MRI and to determine tumor invasion borders in concordance with histological, immunohistochemical, and molecular genetic data from highly perfused enhancing tumor core, perifocal infiltrative edema zone, and normal-appearing peritumoral white matter along the surgical approach.

## 2. Materials and Methods

### 2.1. Enrollment Criteria

The project was approved by the local ethical committee (protocol 02/2019). All patients signed informed consent forms. The study enrolled 50 grade 3–4 glioma patients aged 30 to 72 years (Table 1); all tumors were unilateral. No one underwent neurosurgical operations prior to enrollment. All patients were operated on in N.N. Burdenko National Medical Research Center of Neurosurgery from 2019 to 2021; thus, glioma grading was firstly based on the 2016 WHO classification [42] but then was additionally updated according to the 2021 WHO classification (Table 1) [43]. Histopathological diagnosis was made by two experienced (10 years) neuropathologists. The enrollment criteria also included a Karnofsky score of not less than 80. Postoperative treatment included radiotherapy and chemotherapy.

All 50 patients underwent tumor removal with neuronavigation-guided biopsy. The following regions were sampled: highly perfused enhancing tumor core, perifocal infiltrative edema zone, and normal-appearing peritumoral white matter along the surgical approach.

### 2.2. Magnetic Resonance Imaging

All patients were scanned 1–3 days prior to surgery on a 3.0 T MR scanner (Signa HDxt, GE, USA) with an 8-channel head coil. The protocol included structural images, DKI, and ASL perfusion.

Conventional MRI. The study employed the following sequences: axial T2 FSE (slice thickness 5 mm, gap 1 mm), axial T2-FLAIR (slice thickness 5 mm, gap 1 mm), sagittal T2-FLAIR CUBE (slice thickness 1.4 mm), DWI ASSET (slice thickness 5 mm, gap 1 mm), 1 × 1 × 1 mm isotropic voxel T1 FSPGR before and after contrast enhancement, axial T1 FSE (slice thickness 5 mm, gap 1 mm) before and after contrast enhancement.

Diffusion kurtosis imaging was performed with SE-EPI pulse sequence, three values of diffusion b-factor (0, 1000, 2500 s/mm^2^), and 60 diffusion gradient directions for every nonzero b-factor value, TR = 10,000 ms, TE = 103.4 ms, FOV = 240 × 240 mm, slice thickness 2.5 mm, zero gap, NEX = 1, axial scanning plane. The time of DKI acquisition was 22 min.

ASL-perfusion. Brain perfusion maps were obtained during 3D pCASL data processing. pCASL sequence had the following parameters: 3D FSE, 8-lead helical entire brain volume scan with subsequent reformatting, slice thickness 4 mm, FOV 240 × 240 mm, matrix 128 × 128, ZIP 512, TR 4717 ms; TE 9.8 ms; NEX 3; post-marking delay (PLD) 1525 ms, pixel bandwidth 976.6 Hz/pixel. Scan duration was 4 min 30 s.

### 2.3. Image Processing

The following software was used for DKI postprocessing: Mathworks Matlab R2015b (http://www.mathworks.com/, accessed on 15 November 2021), ExploreDTI 4.8.6 (http://www.exploredti.com/, accessed on 15 November 2021), FMRIB FSL 5.0 (http://fsl.fmrib.ox.ac.uk/, accessed on 15 November 2021).

In the selected regions of interest (ROIs), the following values were measured: mean diffusivity (MD), fractional anisotropy (FA), mean kurtosis (MK), axial kurtosis (AK), radial kurtosis (RK), kurtosis anisotropy (KA) as well as white matter tract integrity parameters: axonal water fraction (AWF), axial extra-axonal space diffusivity (AxEAD), axial intra-axonal space diffusivity (AxIAD), radial extra-axonal space diffusivity (RadEAD), radial intra-axonal space diffusivity (RadIAD), TORTuosity of extra-axonal space (TORT) [22,24].

DKI parametric maps were calculated in Matlab, where diffusion and kurtosis tensors were calculated by the least square method [44]. Voxels with MK less than zero or higher than 3 were deleted from all kurtosis parametric maps. Co-registration of parametric maps to T1 FSPGR was performed automatically in FSL FLIRT (https://fsl.fmrib.ox.ac.uk, accessed on 15 November 2021) by means of affine transformations. If the images had significant automatic co-registration errors, they were co-registered manually in 3D Slicer software (https://www.slicer.org, accessed on 15 November 2021). In this case, the affine transformation matrix was calculated on the basis of a 12-landmark manual co-registration performed by an experienced radiologist.

ASL-perfusion CBF maps were calculated in GE ReadyView 11.3-14.6 software (GE Healthcare) (http://www.gehealthcare.com/, accessed on 15 November 2021). In selected regions of interest, mean CBF in tumor and normal-appearing white matter were obtained.

In all cases, CBF maps were fused with structural images (T2, T2-FLAIR CUBE, enhanced T1 FSPGR) in NeuroRegistration software (GE Healthcare).

To obtain quantitative values in the same ROIs, all parametric images were co-registered.

Two experienced neuroradiologists (10 and 20 years of experience) manually selected 4 ROIs to obtain quantitative data (Figure 1):ROI1—the highest perfusion focus in enhancing tumor part according to the fusion of perfusion maps and post-contrast T1 FSPGR. As CBF measured by ASL-perfusion has a correlation with tumor grade and the Ki-67 labeling index (LI), this site was regarded as the most malignant [45];ROI2—perifocal infiltrative edema zone according to the fusion of T2-FLAIR CUBE and post-contrast T1 FSPGR (T2-FLAIR CUBE hyperintense signal without signs of pathological contrast enhancement);ROI3—normal-appearing (intact on conventional MRI) white matter along surgical approach according to T2-FLAIR CUBE images (no pathological changes of MR signal);ROI4—contralateral (intact on conventional MRI) unaffected hemisphere normal-appearing white matter (centrum semiovale).

The area of selected ROIs was 44 mm^2^.

All ROIs were projected on parametric maps. Only mean values for every ROI were used for statistical analysis.

### 2.4. Surgery and Biopsy Sampling

Prior to surgery, locations of ROI1, ROI2, and ROI3 were agreed upon and checked by a radiologist and a neurosurgeon with the absolute exclusion of eloquent brain cortex areas and white matter tracts. Then, MR data were loaded into the neuronavigation system.

All 50 patients underwent tumor removal with standard surgical instruments and neuronavigation system application without retractors and endoscopic equipment. Minimal brain shift during biopsy was achieved with the burr hole technique. Prior to tumor removal, tissue samples were obtained from the zones corresponding to ROI1, ROI2, and ROI3. Neuronavigation assistance was performed with Fiagon frameless electromagnetic system (Fiagon GmbH, Hennigsdorf, Germany).

### 2.5. Morphological, Immunohistochemical, and Molecular Genetic Studies of Gliomas

Immunohistochemical analysis was performed in acquired tissue samples from ROI1 (n = 50), ROI2 (n = 50), and ROI3 (n = 49) to determine Ki-67 proliferation activity labeling index (LI) and Bcl-2 antiapoptotic marker expression activity (EA). The molecular genetic analysis included IDH1 and IDH2 mutational status (mutIDH/wtIDH) assessment—Table 1. Of 37 glioblastoma patients, CD133 stem cell marker EA was determined in 35 patients (2 cases of 37 could not be studied for CD133 EA due to technical reasons).

### 2.6. Statistical Analysis

For every ROI, all quantitative values (DKI—MD, FA, MK, AK, RK, KA, AWF, AxEAD, AxIAD, RadEAD, RadIAD, TORT; ASL-perfusion—CBF) were calculated. Values in the following ROIs were compared:ROI1 (enhancing tumor core with highest CBF) and ROI2 (perifocal infiltrative edema zone);ROI2 (perifocal infiltrative edema zone) and ROI3 (normal-appearing peritumoral white matter along surgical approach);ROI3 (normal-appearing peritumoral white matter along surgical approach) and ROI4 (contralateral unaffected hemisphere normal-appearing white matter).

The Mann-Whitney test and ROC analysis with bootstrap replications were done in R-project software (https://www.r-project.org/, accessed on 15 November 2021) to compare the values. The differences were considered significant with *p* < 0.001.

Spearman rank correlation coefficient was calculated between neuroimaging (DKI and ASL) values and immunohistochemical markers (Ki-67 LI, Bcl-2, and CD133 EA for glioblastomas). Results were considered statistically significant at *p* < 0.05. In addition, we assessed a 95% confidence interval of established correlations with the RVAideMemoire R-project library.

## 3. Results

According to histological data, 37 of 50 patients were diagnosed with glioblastoma (grade IV): seven patients with anaplastic oligodendroglioma (grade III) and six patients with anaplastic astrocytoma (grade III) (Table 1). Thirty-seven tumors were IDH1-wildtype (36 glioblastomas, 1 anaplastic astrocytoma (2016), or glioblastoma (2021), 13 tumors were IDH1-mutant (1 glioblastoma (2016), or astrocytoma, grade IV (2021), 5 anaplastic astrocytomas, 7 anaplastic oligodendrogliomas).

Several radiological characteristics according to the VASARI (Visually Accessible Rembrandt Images) feature set are represented in Table 2.

Quantitative DKI and CBF data analysis yielded mean values (Appendix A), which were compared between ROI1, ROI2, ROI3, and ROI4 in the combined group and in the glioblastoma group (Figure 2). As depicted in Figure 2, the tendencies of DKI and CBF changes were similar in the two groups.

Results of quantitative MRI data comparisons between ROIs (*p*-values) are represented in Appendix A.

The highest ASL perfusion CBF values were detected in the enhancing tumor core in both combined and glioblastoma groups (*p* < 0.001). No significant CBF differences between ROIs 2, 3, and 4 were detected.

Significant differences of AK, AxIAD, and RadIAD were detected in ROI2 compared to ROI1 (*p* < 0.001) in both combined and glioblastoma groups.

Significant differences in DKI values were detected between ROI3 (normal-appearing peritumoral white matter along surgical approach) and ROI2 (perifocal infiltrative edema) in the combined patient group (MD, FA, MK, RK, KA, AWF, AxEAD, AxIAD, RadEAD, TORT) and in the glioblastoma group (MD, FA, MK, RK, KA, AWF, RadEAD, TORT).

In addition, significant differences (*p* < 0.001) were detected between ROI4 (contralateral unaffected hemisphere normal-appearing white matter) and ROI3 (normal-appearing peritumoral white matter along surgical approach) in the combined patient group (MD, FA, MK, RK, KA, AWF, AxIAD) and in the glioblastoma group (FA, MK, RK, KA, AWF, AxIAD).

ROC analysis showed the diagnostic significance of CBF for distinguishing highly perfused enhancing tumor core with damaged blood-brain barrier and contrast enhancement from perifocal infiltrative edema in high-grade gliomas. CBF values were significantly higher in ROI1 than in ROI2 (perifocal infiltrative edema zone), which indicates a more pronounced neoangiogenesis in tumor stroma (Table 3).

In the combined patient group, ASL perfusion values were the most informative for the localization of high-grade glioma zones with the highest malignancy compared to the perifocal infiltrative edema zone. Less informative were RadIAD, AxIAD, and AK in both groups and AxEAD in the glioblastoma group (Table 3).

According to the ROC analysis, the following parameters were the most informative for tumor border detection (comparison between ROI2 and ROI3): AWF, MK, RK, FA, MD, KA, TORT, RadEAD in the combined group; AWF, MK, RK, FA, MD, KA, TORT in the glioblastoma group (Table 3).

The most reliable marker for differentiation between possible tumor infiltration of normal-appearing peritumoral white matter (ROI3) and contralateral unaffected hemisphere white matter (ROI4) were the following ones: MK, RK, AWF, KA, FA in the combined group, MK, RK, KA, AWF in the glioblastoma group (Table 3).

We established the following significant (*p* < 0.05) correlations of CBF and DKI with Ki-67 and Bcl-2 in the combined glioma patient group (n = 50): in ROI1 (highly perfused enhancing tumor part) Ki67 LI—CBF (r_s_ = 0.363), Ki67 LI—AWF (r_s_ = 0.343), Ki67 LI—RK (r_s_ = 0.324); in ROI2 (perifocal infiltrative edema zone) Bcl2—MK (r_s_ = −0.444), Bcl2—RK (r_s_ = −0.427). The following significant (*p* < 0.05) correlations were shown in the glioblastoma group (n = 37): in ROI1 (highly perfused enhancing tumor part) Bcl2 EA—AWF (r_s_ = −0.504), Bcl2 EA—AxIAD (r_s_ = −0.385), Bcl2 EA—FA (r_s_ = −0.476), Bcl2 EA—MD (r_s_ = 0.421), Bcl2 EA—RadIAD (r_s_ = 0.444), Bcl2 EA—TORT (r_s_ = −0.501), in ROI2 (perifocal infiltrative edema zone) Ki67 LI—AK (r_s_ = −0.434), Ki67 LI—MK (r_s_ = −0.381), Bcl2 EA—AWF (r_s_ = −0.448), Bcl2 EA—MK (r_s_ = −0.522), Bcl2 EA—RK (r_s_ = −0.497). There were no correlations detected in ROI3 (normal-appearing peritumoral white matter).

In 100% of non-enhancing perifocal infiltrative edema zone tissue samples, we detected tumor cells with Ki-67 LI and Bcl-2 EA lower than in the contrast-enhancing tumor area (Figure 3).

In all 35 tissue samples of non-enhancing perifocal T2- and T2-FLAIR-hyperintensive zone in the glioblastoma group, CD133 marker expression was revealed. Six of thirty-seven samples of peritumoral normal-appearing white matter along the surgical approach contained tumor cell infiltration, and 8 of 35 peritumoral normal-appearing white matter samples showed positive CD133 expression.

## 4. Discussion

Assessment of infiltrative high-grade glioma tumor part spread in zones with intact blood-brain barrier was the main task of our study. It was achieved with MRI biomarkers—a broad spectrum of DKI values as well as CBF obtained with pCASL perfusion. Those values were measured in highly perfused enhancing tumor core, perifocal infiltrative edema zone, in normal-appearing white matter of the affected hemisphere as well as in unaffected contralateral hemisphere white matter. Parametric MRI mapping was accompanied by biopsy sampling from the corresponding ROIs of the affected hemisphere with immunohistochemical and molecular genetic testing.

All samples obtained from the non-enhancing perifocal infiltrative edema zone were infiltrated with tumor cells. Ki-67 LI and Bcl-2 EA in these tissues were lower than in samples from highly perfused enhancing tumor cores. All glioblastoma tissue samples tested for CD133 (35 of 37) were CD133-positive. Our results support former studies which demonstrated infiltration of high-grade glioma perifocal vasogenic edema by tumor cells [1,38].

High-grade glioma tumor cells may be detected at a distance from the tumor core [4]. Biopsies performed in our study proved this fact as tissue samples from normal-appearing peritumoral brain tissue were infiltrated by tumor cells in 6 cases of 37, while CD133 expression was positive in 8 of 35 samples.

DKI is known to reflect white matter microstructure more accurately than other quantitative neuroimaging methods. We proved the significance of DKI values with histopathological data, proliferative and antiapoptotic activity as well as CD133 glioma stem cell marker studied in samples acquired during guided biopsy from regions corresponding to those selected on MRI.

We revealed significant differences in the following DKI parameters between the unaffected contralateral hemisphere (with a shift towards normal values) and normal-appearing peritumoral white matter (suggestive of possible tumor cell infiltration): in the combined group of patients (MD, FA, MK, RK, KA, AWF, AxIAD) and in the glioblastoma group (FA, MK, RK, KA, AWF, AxIAD). Normal DKI values seem to be a sign of the possible absence of tumor cells in the unaffected contralateral hemisphere. Subtle changes mentioned above cannot be detected on routine MRI but are revealed on DKI, which reflects brain tissue microstructure.

Our study showed high sensitivity and specificity of kurtosis parameters as well as FA and MD in differentiating perifocal infiltrative edema zone and normal-appearing peritumoral white matter, supported by significant correlations with Ki-67 LI and Bcl-2 EA.

Qiu J. et al., 2022, conducted a DKI study that enrolled 40 grade 3 and 4 glioma patients [46]. They showed correlations of MK, AK, and RK with Ki-67 LI in solid tumor part and significant differences of these DKI parameters between perifocal edema zones of grade 3 and 4 gliomas. Nevertheless, no correlations were found between the studied parameters and Ki-67 LI in perifocal edema. The authors selected 10 ROIs in each region (solid tumor and edema zone). No tissue samples were obtained from the perifocal edema and normal-appearing white matter along the surgical approach. CBF-based detection of the most malignant regions was not performed either. On the contrary, our study included such ROIs in highly perfused enhancing tumor cores as such tumor parts were proven to be the most malignant [45,47].

According to our results, FA decreased in highly perfused enhancing tumor core and perifocal infiltrative edema zone but increased in normal-appearing peritumoral white matter and was significantly higher in the contralateral hemisphere. Our findings correlate with data from other studies, which indicate FA decrease in perifocal edema independently of its origin [5]. Other authors claim that the presence of glioma cells in the edema zone adds some “organization” to white matter leading to a less significant FA drop compared to pure vasogenic edema [48]. In our opinion, FA is a relatively reliable parameter reflecting white matter integrity and always decreases during the disruption of brain tracts. Kurtosis anisotropy (KA) is a parameter that demonstrates tissue anisotropy in complex white matter fiber intersection zones underestimated by FA [49,50]. In our study, KA was as reliable as FA.

As we demonstrated, parameters reflecting brain matter structure do differ between highly-perfused enhancing tumor core with damaged blood-brain barrier and perifocal infiltrative edema zone with an intact blood-brain barrier but not so strongly, as in both cases, brain structure disorganization is present. At the same time, significant differences in studied values between perifocal vasogenic edema infiltrated by tumor cells, normal-appearing peritumoral white matter, and unaffected contralateral hemisphere white matter indicate that tissue structure partially persists in the peritumoral zone and remains normal contralaterally. These changes in quantitative values point to the possible presence of tumor cells in radiologically normal-appearing peritumoral white matter, which was proven by biopsy results.

Many published studies describe the assessment of DKI in glioma grading [25,26,27,51,52] and show the high capacity of the method [32,34,35].

Assessment of the perifocal zone with DKI provides an opportunity to acquire more precise data (as compared to DWI and DTI) to differentiate infiltrative edema from purely vasogenic edema, e.g., in glioblastomas and metastases, respectively. Infiltrated brain matter has a more complex structure with the presence of tumor cells, gliosis, edema, and neoformed vasculature leading to an increase of mean kurtosis, while pure edema possesses lower MK values [33,36,38].

Delgado A. et al., 2017, enrolled 35 patients with astrocytomas and oligodendrogliomas (grades II and III) and studied DKI parameters in peritumoral normal-appearing white matter compared to the unaffected contralateral hemisphere, as well as defined tumor grades and histological subtypes [35]. The authors found a significant difference in DKI parameters between peritumoral white matter and the contralateral hemisphere, reflecting changes invisible on conventional MRI. MK, RK, and FA were increased in peritumoral white matter and in the contralateral hemisphere compared to the solid tumor part, while MD was decreased. In concordance, we also demonstrated MK decrease in tumor and MK rise in peritumoral normal-appearing white matter and unaffected contralateral hemisphere white matter.

Tan Y. et al., 2015, studied MK, RK, AK, as well as FA and MD in solid tumor part and perifocal edema for high-grade glioma and solitary metastasis differentiation. There were no differences between solid tumor parts. This fact was explained as a similar way of disorganized white matter substitution by tumor masses in both cases. In perifocal edema, MK, RK, and AK were significantly higher in high-grade gliomas than in metastases. MK reflects tumor tissue complexity and perifocal edema structure in gliomas is more compound than in metastases due to infiltration [36]. Directional kurtosis (AK and RK) yields specific data concerning diffusion direction. These parameters, according to Tan Y. et al., 2015, are higher in perifocal edema in astrocytomas as tumor cells restrict diffusion in both radial and axial directions. Additionally, the sensitivity and specificity of AK were higher than those of RK, possibly indicating more pronounced axial rather than radial diffusion restriction. In our study, RK, as well as MK, had the highest sensitivity and specificity for comparison between ROIs in perifocal edema and peritumoral white matter or contralateral white matter, while AK did not show diagnostic significance. Besides, Tan Y. et al., 2015, did not perform histological verification of perifocal edema changes [36].

We studied DKI parameters reflecting white matter microstructure—AWF, AxEAD, AxIAD, RadEAD, RadIAD. Fieremans et al., 2011, and Grossman et al., 2015, identified AxEAD and RadEAD as potential markers of the following extra-axonal processes: inflammation, gliosis, and, generally, demyelination [22,24]. AWF is a potential marker of axonal loss and expresses the relation of intra-axonal water volume to the sum of extra- and intra-axonal water volumes, except myelin. TORT may be regarded as a demyelination predictor. These parameters are not informative for gray matter because gray matter does not possess two-component model features assuming the presence of an intra-axonal compartment (where axons are regarded as “infinitely long cylinders”) and an extra-axonal compartment with no diffusion interchange, while myelin is considered water-impermeable [24]. Nevertheless, our results revealed possibilities of these parameters to show changes in tissue structure when white matter and tumor are compared. White matter microstructure in solid tumors is distorted due to high tissue heterogeneity. In perifocal edema, white matter parameters are altered due to a combination of vasogenic edema and tumor infiltration. However, beyond MR-signal changes, pathological DKI alteration may also be explained by tumor cell spread. In our study, ROC analysis showed the highest significance of the AWF parameter, which was reduced in both highly perfused enhancing tumor core and perifocal infiltrative edema zone and increased in peritumoral normal-appearing white matter and (even more) in the contralateral hemisphere. This fact indicates that axons are less damaged in the peritumoral zone and remain intact in the contralateral hemisphere. AWF allows differentiating between perifocal infiltrative edema and normal-appearing peritumoral white matter. It shows significant differences with unaffected contralateral hemisphere white matter, thus having higher sensitivity and specificity than other values.

Formerly, Akbari H. et al., 2016, performed multiparametric glioblastoma perifocal zone analysis employing T1 before and after contrast enhancement, T2, T2 FLAIR, DTI, dynamic susceptibility contrast perfusion and machine learning. The authors demonstrated perifocal heterogeneity and tumor infiltration patterns which predict tumor spread direction but did not perform perifocal infiltration zone tissue sampling. The proposed model allowed anticipating foci with high tumor relapse probability, but there were cases of false-positive prediction. This work deserves interest, has perspective, and needs further investigation [37].

Kim J.Y. et al., 2019, studied the non-enhancing perifocal zone in 83 glioblastoma patients with a radiomics model combining FA and normalized DSC perfusion CBV. The applied model showed a high prognostic level for 6-month tumor progression. The authors did not obtain molecular data. They also underlined that verifying radiomics by studying the entire corresponding tumor tissue is currently unavailable [39].

pCASL perfusion imaging applied in our study showed that CBF was significantly higher in highly perfused enhancing tumor core than in perifocal infiltrative edema, which indicates prominent neoangiogenesis in tumor stroma. No significant CBF increase was detected in any zone beyond enhancing tissue. However, our study established a significant correlation between CBF and Ki-67 LI (r_s_ = 0.363) in highly perfused enhancing tumor cores in the combined patient group (n = 50). This generally supports our previous studies. We consider that biopsy should be taken from the highest perfusion focus defined by ASL (pCASL) perfusion [9,45,47].

We established various significant correlations between immunohistochemical and DKI values. Particularly interesting are correlations observed in non-enhancing perifocal edema as they confirmed brain matter alterations, presence, and prominence of edema infiltration by tumor cells. 

Also, we obtained quantitative MRI data and biopsy samples from the same regions not only in highly perfused tumor core and perifocal infiltrative edema but also in the normal-appearing white matter along the surgical approach. Such methodology is unique, and we found no analogs. Only several studies represented matching of MRI and histopathological data beyond the radiological tumor border [40,41,53,54]. These works recruited small patient groups predominantly with low-grade gliomas or combined low- and high-grade tumors. At the same time, postmortem studies in high-grade gliomas demonstrated the presence of tumor cells at a high distance from the main tumor lesion [4,55].

In the work of Guo J. et al., 2012, needle biopsies were taken from the perifocal infiltrative zone and intact peritumoral brain matter of 18 grade 2–4 glioma patients under MRI control. Cho/NAA ratio had a prognostic force for tumor cell detection with different cutoff values for low- and high-grade gliomas. The authors showed a correlation of Cho/NAA with MIB-1 and CD34 marker expression, as well as with tumor infiltration. Cho/NAA ratio was demonstrated to be unique for glial tumor border detection [40].

Zetterling M. et al., 2016, performed 5 grade 2 and 3 gliomas en bloc resections and matched IDH1 immunohistochemical study results with changes on T2 and T2-FLAIR images. Tumor cells were detected along white matter tracts beyond radiological tumor borders in all cases. The main conclusion stated that T2-FLAIR was insufficiently sensitive to tumor infiltration detection [41].

In our work, CD133 glioma stem cell marker expression was determined in glioblastoma patients [56]. On the glioblastoma periphery, CD133-positive cells are responsible for active brain invasion, further tumor spread, and involvement of new brain structures [57]. Comparative studies of CD133 expression in different morphological and topographic regions have never been done previously. Such comparison, in our opinion, allows the acquisition of novel data concerning tumor features that reflect patterns of tumor stem cell spread and distribution. We studied CD133 in highly perfused enhancing tumor core, in perifocal infiltrative edema zone, and normal-appearing peritumoral brain tissue. The study of CD133 distribution in radiologically normal-appearing brain tissue is a unique part of our work as it allows assessment of glioma stem cell presence and dissemination in normal-appearing brain matter.

CD133 was detected in all 35 glioblastoma samples. This fact confirms the statement that the perifocal edema zone in malignant intra-axial tumors is infiltrated by tumor cells. Moreover, in eight cases, the normal-appearing peritumoral white matter was CD133-positive. At the same time, changes in DKI parameters in this zone were revealed in other studied patients. We assume that this may be due to possible glioblastoma cell infiltration of peritumoral radiologically intact white matter.

Our study has several limitations. The first one is the selection of small ROIs for quantitative data acquisition. Currently, spreading radiomics and pathological tissue segmentation methods, to some extent, are targeted at a bias-effect reduction. Nevertheless, our methodology with a sampling of small tissue fragments precisely implies ROI selection. We assumed that obtaining tissue samples as well as quantitative MRI data from the same regions would reflect tissue conditions more reliably. However, the tumor may possibly spread and infiltrate brain tissues in the opposite direction from sometimes the only possible biopsy trajectory along the surgical approach. Glioma infiltration is known to be uneven in relation to the main tumor mass, while the prevalence of medial growth pattern is also reported [40,58,59]. The selection of ROIs in our studies was performed with the complete exclusion of eloquent brain cortex areas and white matter tracts.

The second limitation is due to the absence of logistic regression-based calculations in result processing. Logistic regression could help to choose the best combination of parameters illustrating alteration or normalization of brain structure, so we will certainly use this opportunity in the future.

It should be noted that diffusion kurtosis parameters are still remaining the subject of research works but are not used routinely due to the long process of data acquisition, labor-intensive and time-consuming post-processing. At the same time, we think that the application of a complex of MRI parameters for brain matter integrity assessment, not a single parameter (e.g., mean kurtosis), provides the chance to understand the comprehensive structural changes.

Another limitation of our study is the absence of post-operative, post-irradiation, and chemotherapy-accompanying follow-up MRI analysis. However, we have been monitoring several patients, and we will certainly carry on with the follow-up.

Artificial intelligence application is also a perspective direction for such studies. We are going to study AI opportunities when sufficient data is present.

## 5. Conclusions

We determined diffusion kurtosis biomarkers which may be used for the detection of high-grade glioma invasion borders at the periphery of enhancement and showed correlates with immunohistochemical data. Glioma-enhancing core perfusion has a direct relation with proliferative activity. Our results may be useful for future plotting of individual tumor invasion maps for brain glioma surgery or radiotherapy planning.

Presented methodology of high-grade glioma and normal-appearing white matter multiparametric imaging was used in tumor infiltration zone border detection for the first time.

## Figures and Tables

**Figure 1 cancers-15-02760-f001:**
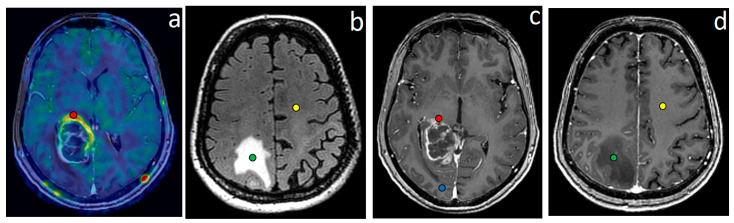
Segmentation of ROIs on MRI of a glioblastoma patient for analysis of DKI and ASL parameters (in both affected and contralateral hemispheres) and biopsy planning (in affected hemisphere): (**a**)—fused CBF map and enhanced T1 FSPGR, (**b**)—T2-FLAIR CUBE, (**c**,**d**)—enhanced T1 FSPGR. Red—ROI1, green—ROI2, blue—ROI3, yellow—ROI4.

**Figure 2 cancers-15-02760-f002:**
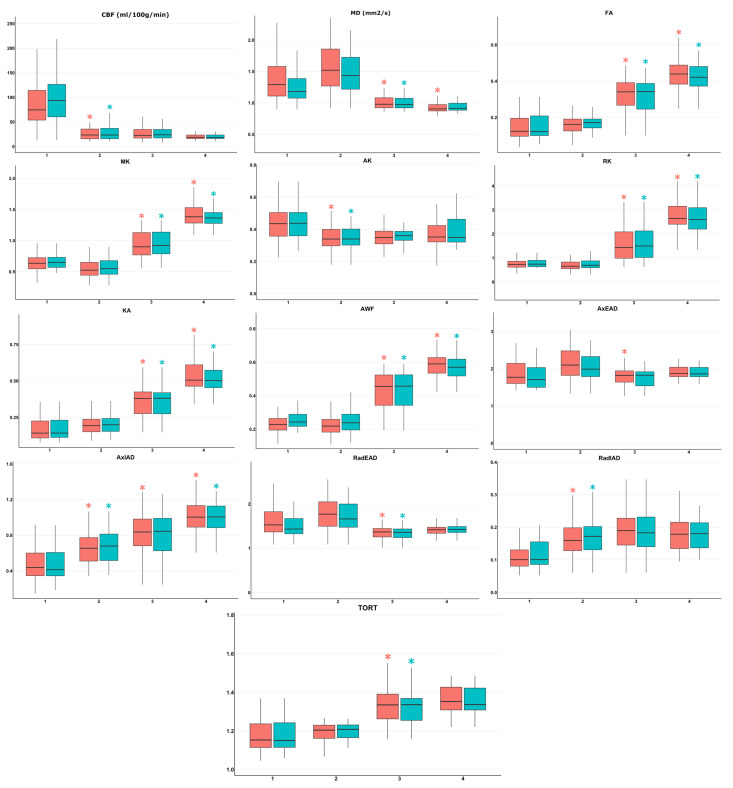
Mean quantitative values of DKI and ASL-perfusion for the combined group of patients (n = 50, red boxes and asterisks) and for the glioblastoma group (n = 37, blue boxes and asterisks) with significant differences highlighted by asterisks (*p* < 0.001) between ROI1 (highly perfused enhancing tumor core), ROI2 (perifocal infiltrative edema zone), ROI3 (normal-appearing peritumoral white matter along surgical approach), ROI4 (contralateral unaffected hemisphere normal-appearing white matter). Legend: 1—ROI1, 2—ROI2, 3—ROI3, 4—ROI4, *—*p* < 0.001, horizontal lines within the boxes represent medians, lower and upper hinges correspond to the first and third quartiles, whiskers extend at 1.5 * interquartile range from the hinges.

**Figure 3 cancers-15-02760-f003:**
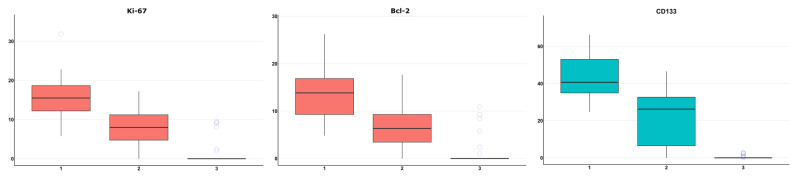
Dependence of Ki-67 LI (n = 50), Bcl-2 EA (n = 50) (ROI1 n = 50, ROI2 and ROI3 n = 49, combined group, red boxes), and CD133 EA (in the glioblastoma group, blue boxes, n = 35) on the ROI. Horizontal lines within the boxes represent medians, lower and upper hinges correspond to the first and third quartiles, and whiskers extend at 1.5 * interquartile range from the hinges.

**Table 1 cancers-15-02760-t001:** Histological and molecular genetic data for glioma patients, grade III–IV (n = 50) (WHO classifications, 2016, 2021).

Diagnosis, Grade WHO 2016	Diagnosis, Grade WHO 2021	MGMT Methylation	IDH1 Mutation	IDH2 Mutation	Number of Patients
Anaplastic astrocytoma III	Astrocytoma 3	+	+	−	4
Anaplastic astrocytoma III	Astrocytoma 3	−	+	−	1
Anaplastic astrocytoma III	Glioblastoma 4	−	−	−	1
Anaplastic oligodendroglioma III	Oligodendroglioma, 3	+	+	−	4
Anaplastic oligodendroglioma III	Oligodendroglioma, 3	−	+	−	3
Glioblastoma IV	Astrocytoma 4	−	+	−	1
Glioblastoma IV	Glioblastoma 4	+	−	−	10
Glioblastoma IV	Glioblastoma 4	−	−	−	26
Total					50

**Table 2 cancers-15-02760-t002:** Selected radiological features of gliomas according to the VASARI set.

Criterion	Feature
Enhancement pattern	None: 1 patient—anaplastic astrocytoma
	Multinodular: 4 patients: - Anaplastic astrocytoma—3 patients - Anaplastic oligodendroglioma—1 patient
	Diffuse: 6 patients: - Anaplastic oligodendroglioma—4 patients - Glioblastoma—2 patients
	Ring-like: 40 patients: - Glioblastoma—35 patients - Anaplastic oligodendroglioma—2 patients - Anaplastic astrocytoma—3 patients
Necrosis	Of 50 patients, present in 42, absent in 8
Hemorrhage	Of 50 patients, present in 44, absent in 6

**Table 3 cancers-15-02760-t003:** ROC-analysis of quantitative MRI parameter differences between ROI1 and ROI2, ROI2 and ROI3, ROI3, and ROI4 in the combined glioma group and in the glioblastoma group. Significant differences between ROIs are highlighted in bold.

ROI1 vs. ROI2 (Combined)					ROI1 vs. ROI2 (Glioblastoma)			
	AUC	Cutoff	Specificity	Sensitivity	AUC	Cutoff	Specificity	Sensitivity
CBF	**0.89**	**50.13**	**0.81**	**0.93**	**0.92**	**50.9**	**0.9**	**0.9**
RadIAD	**0.78**	**0.11**	**0.66**	**0.89**	**0.77**	**0.5**	**0.69**	**0.79**
AxIAD	**0.76**	**0.51**	**0.71**	**0.76**	**0.77**	**0.11**	**0.66**	**0.9**
AK	**0.74**	**0.41**	**0.63**	**0.76**	**0.77**	**0.41**	**0.66**	**0.76**
AxEAD	0.69	1.92	0.63	0.66	0.71	1.78	0.66	0.76
MK	0.67	0.57	0.71	0.63	0.67	0.57	0.76	0.59
KA	0.66	0.15	0.55	0.82	0.67	1.3	0.66	0.66
MD	0.65	1.4	0.63	0.63	0.66	0.15	0.55	0.86
RadEAD	0.64	1.58	0.63	0.66	0.65	1.48	0.59	0.76
TORT	0.63	1.16	0.58	0.74	0.62	1.16	0.59	0.76
FA	0.61	0.13	0.55	0.74	0.61	0.14	0.55	0.76
RK	0.6	0.73	0.53	0.71	0.59	0.72	0.62	0.62
AWF	0.56	0.22	0.55	0.55	0.56	0.24	0.55	0.55
**ROI2 vs. ROI3 (Combined)**					**ROI2 vs. ROI3 (Glioblastoma)**			
	**AUC**	**Cutoff**	**Specificity**	**Sensitivity**	**AUC**	**Cutoff**	**Specificity**	**Sensitivity**
AWF	**0.93**	**0.31**	**0.92**	**0.82**	**0.91**	**0.31**	**0.9**	**0.8**
MK	**0.93**	**0.72**	**0.92**	**0.79**	**0.9**	**0.77**	**0.93**	**0.77**
RK	**0.92**	**0.9**	**0.84**	**0.82**	**0.9**	**1**	**0.86**	**0.77**
FA	**0.9**	**0.21**	**0.87**	**0.85**	**0.88**	**0.21**	**0.83**	**0.87**
MD	**0.9**	**1.2**	**0.82**	**0.87**	**0.88**	**1.11**	**0.86**	**0.83**
KA	**0.87**	**0.25**	**0.82**	**0.82**	**0.84**	**0.25**	**0.76**	**0.83**
TORT	**0.84**	**1.25**	**0.87**	**0.79**	**0.81**	**1.27**	**0.97**	**0.73**
RadEAD	**0.83**	**1.49**	**0.79**	**0.87**	**0.8**	**1.47**	**0.76**	**0.83**
AxEAD	**0.73**	**1.96**	**0.63**	**0.77**	0.69	1.96	0.55	0.8
AxIAD	**0.72**	**0.72**	**0.68**	**0.72**	0.67	0.82	0.76	0.57
RadIAD	0.6	0.19	0.71	0.51	0.55	0.19	0.66	0.5
AK	0.52	0.35	0.58	0.54	0.53	0.34	0.52	0.67
CBF	0.5	23.92	0.5	0.56	0.51	17.65	0.4	0.72
**ROI3 vs. ROI4 (Combined)**					**ROI3 vs. ROI4 (Glioblastoma)**			
	**AUC**	**Cutoff**	**Specificity**	**Sensitivity**	**AUC**	**Cutoff**	**Specificity**	**Sensitivity**
MK	**0.92**	**1.21**	**0.9**	**0.82**	**0.89**	**1.25**	**0.9**	**0.77**
RK	**0.91**	**2.39**	**0.92**	**0.77**	**0.87**	**2.08**	**0.73**	**0.87**
AWF	**0.87**	**0.53**	**0.79**	**0.77**	**0.87**	**0.43**	**0.8**	**0.83**
KA	**0.87**	**0.44**	**0.79**	**0.85**	**0.83**	**0.53**	**0.77**	**0.7**
FA	**0.81**	**0.4**	**0.79**	**0.72**	**0.79**	**0.4**	**0.83**	**0.67**
MD	**0.75**	**0.94**	**0.69**	**0.69**	0.75	0.93	0.7	0.67
AxIAD	**0.74**	**0.93**	**0.69**	**0.69**	**0.68**	**0.91**	**0.87**	**0.5**
AxEAD	0.64	1.84	0.59	0.67	0.67	1.42	0.73	0.6
RadEAD	0.62	1.42	0.72	0.51	0.64	1.84	0.6	0.63
CBF	0.6	22.3	0.54	0.71	0.64	22.34	0.55	0.73
TORT	0.57	1.37	0.72	0.49	0.56	1.37	0.73	0.43
RadIAD	0.56	0.19	0.51	0.62	0.53	0.19	0.5	0.6
AK	0.55	0.35	0.51	0.56	0.48	0.36	0.5	0.6

## Data Availability

The authors confirm that the data supporting the findings are available within the current article and in appendices. Raw data concerning the results of neuropathological studies are available from the first author (N.E.Z., nzakharova@nsi.ru) upon reasonable request.

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
