# Peer review of "Perifocal Zone of Brain Gliomas: Application of Diffusion Kurtosis and Perfusion MRI Values for Tumor Invasion Border Determination"

_cancers, 2023, doi:10.3390/cancers15102760_

Round 1

Reviewer 1 Report

This is a very nicely designed study addressing an important question regarding the tumour margin and infiltration zone.  The results are interesting and the conclusions valid, suggesting that cellularity markers are more sensitive than perfusion markers. 

The Introduction is well-written and the methods are clear.  However, the study is let down by the presentation of the data. 

Figure 1:  I cannot see ROI 3 (blue) - needs to be added.

Figures 2 and 3: need to be revised.  I would suggest dot plots so that the reader can see individual patient points and can observe the scatter.  I would also suggest adding confidence intervals or standard deviation.

There is also no information on the morphology of the tumour cohort.  A table of basic VASARI criteria would be helpful here - for example, how many tumours were enhancing and how many had non-enhancing components?  How did the authors perform ROIs on tumours where there was no perifocal tumour infiltrative zone (ROI 2)?

Once these points are addresses, i would be happy to recommend publication.

Author Response

Dear reviewer,

we would like to thank you for your effort to evaluate our work. 

We provide replies to all your comments.

Comment 1: Figure 1: I cannot see ROI 3 (blue) – needs to be added.

Response: The ROI 3 was selected in the right occipital lobe (Figure 1, C), the dot has already been placed on the original submitted figure.

Comment 2: Figures 2 and 3: need to be revised. I would suggest dot plots so that the reader can see individual patient points and can observe the scatter. I would also suggest adding confidence intervals or standard deviation.

Response: We considered the proposed methods of figure modification. Unfortunately, if we add too many dots on the plots, these dots will superimpose, and the figures will become overloaded with data thus becoming difficult to understand. Our original figures seem more obvious.

Comment 3: There is also no information on the morphology of the tumour cohort.  A table of basic VASARI criteria would be helpful here – for example, how many tumours were enhancing and how many had non-enhancing components?

Response: We introduced a table that describes the tumors according to VASARI. The VASARI system contains many features and excessive information may overload the paper. Thus, we chose only several criteria (enhancement pattern, necrosis, hemorrhage).

Comment 4: How did the authors perform ROIs on tumours where there was no perifocal tumour infiltrative zone (ROI 2)?

Response: All tumors in our study had perifocal infiltrative edema zone (in all cases, the ROI2 was chosen). The ROI2 was placed at the edge of signal changes on T2WI and T2-FLAIR closer to border with MRI-intact brain matter in all cases regardless of presence or absence of contrast enhancement.

Once again, thank you,

Corresponding author.

Reviewer 2 Report

The present study expand the topic of glioma extension beyond the contrast enhancement portion utilizing ASL and DKI to identify tumor presence.
I have few observation:

- While the selection of ROI on contrast tumor is clear (based on CBF value), the same cannot be said for ROI in peritumoral T2 alteration. Can you please better specify which parameters the authors have used to decide where to draw it?

- Other papers (see 10.1111/jon.12903) have used the whole T2 alteration in response to paper suggesting the removal of the whole tumor alteration (10.1001/jamaoncol.2019.6143). Did any removal of T2 alteration occur? It would be interesting to investigate on the correlation of DKI measures and survival as the suggested paper did.

- Being the tumors diagnosed based on the 2016 WHO classification, it would be good if the authors could update the diagnosis in order to see the difference in DKI values based on molecular alteration (especially for what concerns the IDHmut glioblastoma).

- Table 2 is a little confusing. Could the authors add a space between the different sections of the table and highlight the post relevant values in bold?

- Discussion is too long and difficult to read

Author Response

Dear reviewer,

we would like to express our gratitude for your effort to assess our work. 

We provide response to all your critical remarks.

Comment 1: While the selection of ROI on contrast tumor is clear (based on CBF value), the same cannot be said for ROI in peritumoral T2 alteration. Can you please better specify which parameters the authors have used to decide where to draw it?

Response: The ROI2 was placed at the edge of signal changes on T2WI and T2-FLAIR closer to border with MRI-intact brain matter in all cases regardless of presence or absence of contrast enhancement.

Comment 2: Other papers (see 10.1111/jon.12903) have used the whole T2 alteration in response to paper suggesting the removal of the whole tumor alteration (10.1001/jamaoncol.2019.6143). Did any removal of T2 alteration occur? It would be interesting to investigate on the correlation of DKI measures and survival as the suggested paper did.

Response: The aim of our research was to study structural and hemodynamic alterations in particularly precise tumor regions where biopsy specimens were obtained. Thus, the whole areas of T2 or T2-FLAIR alterations were not considered. The study of postoperative changes and patient survival was not an objective of the current research step, but it will be performed later. The zone of T2-FLAIR changes was included into resection where possible (excluding eloquent areas).

Comment 3: Being the tumors diagnosed based on the 2016 WHO classification, it would be good if the authors could update the diagnosis in order to see the difference in DKI values based on molecular alteration (especially for what concerns the IDHmut glioblastoma).

Response: We updated the table 1 and added the IDH1-status data. Comparison of DKI features of mutant and wildtype tumors will be one of our future goals. Because of a skew towards wildtype gliomas in our sample, the groups are not equal in number of cases, and the statistical results may be biased.

Comment 4: Table 2 is a little confusing. Could the authors add a space between the different sections of the table and highlight the post relevant values in bold?

Response: We added spaces between different table sections and highlighted significant differences in bold.

Comment 5: Discussion is too long and difficult to read.

Response: We tried to make our discussion thorough with extensive comparisons with the results of the other studies. Thus, we had to cite those results in detail, which may seem a bit bulky, but, in our opinion, the discussion part is the most valuable. With your permission, we would like to leave the discussion as is.

Once again, thank you.

Corresponding author.

Round 2

Reviewer 1 Report

The authors have addressed the comments well apart from Figure 2.  The data presentation is completely inappropriate: ROIs are independent so should not be connected by a line.  If there are two many data points, a bar chart could be shown. Mean and standard deviations should be shown on the chart.

Author Response

Dear reviewer,

we have redesigned figures 2 and 3 to represent more statistical information and to make these figures more correct.

Thank you for the remarks.

Reviewer 2 Report

Authors' responses are sufficient for me.

Author Response

Dear reviewer, we thank you for all the remarks!

Round 3

Reviewer 1 Report

I am now satisfied with the change in Chart style.